# Venous puncture wound hemostasis results in a vaulted thrombus structured by locally nucleated platelet aggregates

Sung W. Rhee[1,4], Irina D. Pokrovskaya[2,4], Kelly K. Ball[1], Kenny Ling [3], Yajnesh Vedanaparti [3],
Joshua Cohen [3], Denzel R. D. Cruz[3], Oliver S. Zhao [3], Maria A. Aronova[3], Guofeng Zhang[3],
Jeffrey A. Kamykowski[2], Richard D. Leapman [3] & Brian Storrie [2✉]

Primary hemostasis results in a platelet-rich thrombus that has long been assumed to form a solid plug. Unexpectedly, our 3-dimensional (3D) electron microscopy of mouse jugular vein puncture wounds revealed that the resulting thrombi were structured about localized, nucleated platelet aggregates, pedestals and columns, that produced a vaulted thrombus capped by extravascular platelet adherence. Pedestal and column surfaces were lined by procoagulant platelets. Furthermore, early steps in thrombus assembly were sensitive to $P2Y_{12}$ inhibition and late steps to thrombin inhibition. Based on these results, we propose a Cap and Build, puncture wound paradigm that should have translational implications for bleeding control and hemostasis.

[1] Department of Pharmacology and Toxicology, University of Arkansas for Medical Sciences, Little Rock, AR, USA. [2] Department of Physiology and Cell Biology, University of Arkansas for Medical Sciences, Little Rock, AR, USA. [3] Laboratory of Cellular Imaging and Macromolecular Biophysics, National Institute of Biomedical Imaging and Bioengineering, National Institutes of Health, Bethesda, MD, USA. [4]These authors contributed equally: Sung Rhee and Irina D. Pokrovskaya. ✉email: StorrieBrian@uams.edu

Vascular damage comes in many forms with the puncture wound, be it from a thorn or sharp metal object, among the longest known to human experience. The visible features are blood and the extravascular scab. What happens inside the puncture hole remains invisible to the human eye.

Experimentally, vascular damage has typically been visualized in mouse models, e.g., laser[1–4] and/or ferric chloride treatment [e.g.,[5]] or small needle pricks of arterioles[6] to induce vessel damage. Under these conditions, damage to the endothelial layer lining the vessel wall exposes the underlying collagen-rich adventitia, but often fails to puncture the adventitia to yield an open hole and more than very limited bleeding [for review, see[7–10]]. Efforts to visualize the thrombi have varied from intravital staining imaged by confocal or 2-photon (2P) fluorescence microscopy to ex vivo imaging by the same approaches or at higher resolution by scanning electron microscopy. Interpretation has been dominated by the outcomes seen by intravital microscopy[1,11] in which an initial platelet deposition associated with the damaged vessel wall is followed by α-granule secretion[12] in many of the platelets and cell-surface exposure of p-selectin, an α-granule membrane protein. Subsequent platelet deposition is marked by the accumulation of a less activated outer layer of platelets. In brief, this has given rise to a Core and Shell paradigm of platelet response to vessel damage[6,11], leading to the formation of a self-limiting platelet thrombus and vascular repair. Recently, additional mouse models have been presented in which the adventitia is punctured to create a 75−125[3,13] to 600 μm-diameter wound hole[14]. In the jugular vein case, profuse bleeding occurs, with cessation times varying from less than 60−250 s or more, for the smaller and larger wound diameters, respectively[3,13,14]. Puncture wound results have been interpreted within framework extensions of the Core and Shell paradigm [e.g.,[15,16]]. However, two important aspects of the experimental data[14] suggest that the paradigm may not explain the observations. First, p-selectin exposure as a marker for α-granule secretion and platelet activation is typically concentrated in localized puncta rather than in a well-defined Core. Second, the hemostatic thrombus when viewed at early stages showed a pebbly distribution of platelet aggregates suggestive of nucleated platelet accumulation rather than the smooth layers that would follow from a Core and Shell paradigm. The outgrowth of these aggregates could lead to a vaulted structure rather than a solid platelet plug.

In the Core and Shell paradigm [for review, see [17]], it has been assumed that the formation of a solid, platelet plug is the final result of primary hemostasis, a process in which bleeding cessation is produced in a stepwise manner through platelet adhesion to the exposed vessel wall, subsequent platelet activation and finally plug formation[18]. Taking an initial cue from nucleated platelet aggregate formation, we tested the possibility that thrombus formation within a nominal 300 μm diameter, jugular vein, puncture hole forms a vaulted structure in which platelet activation patterns follow from the underlying structure. To achieve this goal, we visualized the interior and overall structure of the forming puncture wound thrombus fully in 3D at sub-platelet level resolution using serial block face scanning electron microscopy (SBF-SEM) to give a "coarse grain" full thrombus structure supported by more detailed higher resolution wide area transmission electron microscopy (WA-TEM). We adopted in vivo antibody labeling procedures[14] to enable correlative light microscopy. Hence, we could map the distributions of p-selectin, a marker of platelet secretion, and fibrin, in the context of detailed 3D thrombus ultrastructure at 3 nm resolution.

Our raw data consisting of thousands of sequential electron micrographs, revealed a vaulted, platelet-rich thrombus in which bleeding cessation came from capping the hole from the extravascular side rather than filling the hole to form a solid platelet plug. At 5 min post-puncture marked by full bleeding cessation, we found that the forming platelet-rich thrombus had a Swiss cheese-like interior containing vaults that were continuous with the intravascular vessel lumen. Red blood cells (RBCs) that accumulated within the vaults typically had a distorted, nearly polyhedral shape indicative of accumulation under pressure[19]. The surface of the vaults were lined with a multi-platelet thick layer of degranulated platelets, a platelet activation pattern fully consistent with exposed p-selectin staining, and one unexpected by a Core and Shell paradigm. Based upon morphology, the vault-lining of degranulated platelets appeared to contain procoagulant platelets and hence the vaults provide a potential protected surface for coagulation factor activation. Consistent with this suggestion, platelet degranulation was sensitive to the directly acting anti-coagulant drug, dabigatran[20] while extravascular thrombus capping was sensitive to the directly acting P2Y$_{12}$ ADP-receptor inhibitor, cangrelor[21].

We conclude that in a true puncture wound primary hemostasis is structured about the formation of a vaulted platelet-rich thrombus in which platelet activation patterns appear to be a consequence of thrombus geometry. We propose a "Cap and Build" paradigm of primary hemostasis in which localized platelet aggregates are the starting element upon which all subsequent steps in puncture wound thrombus formation builds. In conclusion, we suggest that our mothers' repeated admonitions to not play with the scab have a much stronger basis in fact than once suspected.

## Results

**Experimental system.** As illustrated in cartoon form (Fig. 1a) a 30-gauge needle produces an open hole in the damaged endothelial lining of the jugular vein and exposes the underlying collagen-rich adventitia (Supplementary Fig. 1). As modeled, vessel wall-associated tissue factor is expected to be exposed (for review, see [22,23]) and blood flow is both lateral within the lumen of the vein and outward through the hole. The actual hole formed had a measured diameter of 192 ± 62 μm, 3D SBF-SEM imaging of fixed, plastic-embedded 1 min thrombus preparations (Table 1, n = 4). Mouse jugular vein bleeding times as determined by eye were robust, ~1 min, irrespective of site, (Fig. 1b), and sex independent (Fig. 1c). The forming thrombus was localized in the fixed, excised vessel wall by 2P fluorescence microscopy against the distribution of infused fluorescent antibodies (CD41, red fluorescence, Fig. 1d). Samples were then embedded in plastic (Fig. 1E) and the thrombus block face imaged by SBF-SEM (Fig. 1f).

As shown in Fig. 1f (blue vessel wall, also Supplementary Movie 1), the Z-axis of the stack of 1500−2500 images is perpendicular to flow. Images are oriented with the intravascular side "up" and extravascular side "down" so that flow is from left to right. Even a single, 1 min post-puncture image is sufficient to illustrate that much of early thrombus ultrastructure is organized about, dark gray, platelet aggregates that project inward from the exposed adventitia rimming the puncture hole and "upward" into the intravascular vessel lumen (see Supplementary Movie 1 for a rendered movie view of the overall 1 min thrombus). We refer to the short adventitia-tethered aggregates as "pedestals" and the larger intravascularly projecting platelet aggregates as "columns". As illustrated here and with further examples in Supplementary Fig. 2, all 1 min examples, n = 4, exhibited by EM a visually open puncture cavity, measured diameter of 114 μm ± 68 μm (Table 1), with no entrapment of RBCs within the open cavity.

[*Note*: The EM images in this publication are taken at different raw pixel sizes ranging from 100−3 nm XY (Table 2). Images are

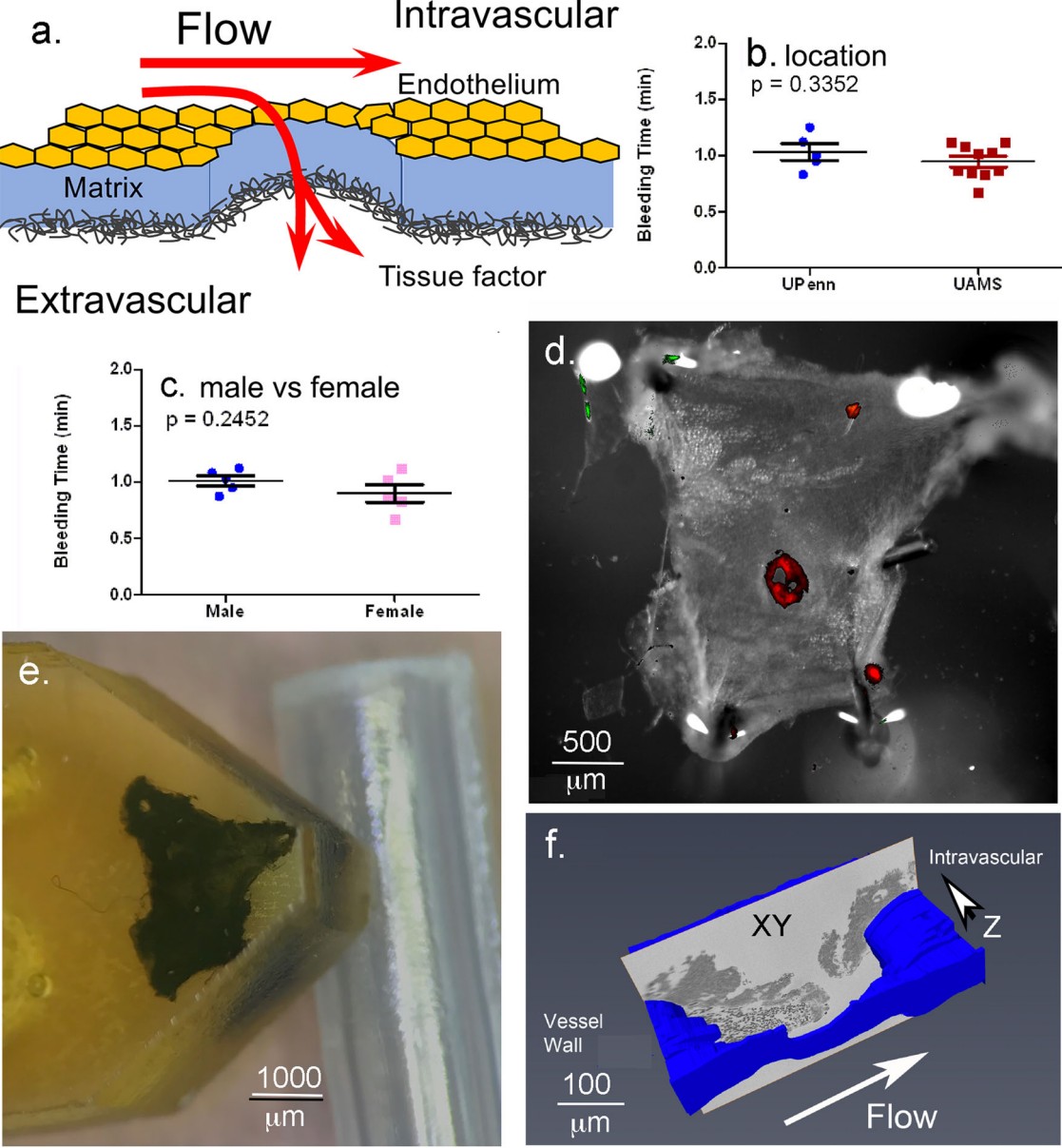

**Fig. 1 Experimental system. a** Schematic of flow and tissue factor exposure in a jugular vein puncture wound. Tissue factor exposure leads to coagulation factor activation, e.g., thrombin generation. **b**, **c** Bleeding time cessation times at the two different locations where punctures were performed and the effect of mouse sex on bleeding times, mean plus and minus standard deviation. Comparative positioning of the thrombus (red fluorescence in (**d**)). Upon opening the jugular vein (**d**) and embedding in plastic (**e**, **f**). Schematic of repeated, SBF-SEM imaging across the puncture wound hole. The resulting image slices are positioned so that flow is from left to right and the direction of progressive block face exposure into the plane of the page.

displayed to show overall structural features of the thrombus and hence are shown at a substantially less resolution than that of the 3, 20, 100 nm pixel size of the original images. The net result is to achieve a "coarse-grained", overall thrombus structure in which the rendered thrombus 3D structure is segmented by platelet state: degranulated, tightly packed, and loosely packed, by reference to example 3 nm $XY$ pixel imaging of single thrombus planes at approximately 25 and 50% thrombus depth (see Supplementary Fig. 3 for an illustrative example)].

**A vaulted, thrombus structures extravascular bleeding cessation, 5 min post-puncture thrombus examples**. We predict that if localized, nucleated aggregate outgrowth is the dominant mode of platelet recruitment to the growing thrombus then bleeding cessation in the jugular model may well be due to sealing the hole

through pedestal outgrowth, rather than filling of the hole. In sum, we predict that puncture wound thrombus growth may well be columnated and hence lead to the formation of a vaulted, extravascularly capped structure.

To test these predictions, we assayed the thrombus structure at 5 min post puncture. As shown by Fig. 2, a typical 5 min jugular puncture wound thrombus proved to be a platelet-rich structure in which zones of degranulated platelets, i.e., highly activated platelets, typically lined the interior surfaces of the vaulted structure (see also Supplementary Fig. 3 for a more detailed example at 3 nm raw pixel size). By 2P immunofluorescence (Fig. 2a, b), fibrin (blue staining) in a typical thrombus was located to the exterior of the structure while mobilized/exposed p-selectin staining (green) was located in multiple, individual puncta in association with the red, CD41 platelet staining. However, because of resolution limits, 2P microscopy yielded

**Table 1 Extent of puncture hole thrombus fill.**

| Sample | Puncture hole diameter (μm) | Unfilled central cavity diameter (μm) | P values (horizontal) | Volume fill (%) |
|---|---|---|---|---|
| 1 min post-puncture | | | | |
| Thrombus 1 | 223 | 65 | | 70 |
| Thrombus 2 | 172 | 113 | | 40 |
| Thrombus 3 | 116 | 74 | | 36 |
| No anti-P-selectin Abs | 229 | 212 | | 18 |
| Mean ± SD, $n = 4$ | 192 ± 62 μm | 114 ± 68 μm | $P = 0.06$ | 26 ± 14% |
| 5 min post-puncture | | | | |
| Thrombus 5 | 167 | 100 | | 40% |
| Thrombus 6 | 166 | 53 | | 68 |
| Thrombus 7 | 189 | 74 | | 61 |
| Thrombus 8 | 198 | 67 | | 66 |
| No anti-P-selectin Abs | 169 | 50 | | 70 |
| Mean ± SD, $n = 5$ | 178 ± 14 μm | 69 ± 20 μm | $P = 0.006$ | 61 ± 12% |
| P values, (vertical) | $P = 0.68$ | $P = 0.28$ | | $P = 0.18$ |

Dimensions and volume fill were determined from 3D renderings of the individual thrombi done in Amira software. The unfilled central cavity is continuous with the vault volume present in the intravascular thrombi crown found 5 min post-puncture. Cavity dimensions decrease with time. SD *standard deviation*. P values were calculated using the Student t-test for either paired data or unpaired data with unequal variance. Calculated P values point to a significant variance in central cavity diameter versus puncture hole diameter, i.e., a numerical indicator of a progressive, localized accumulation of platelets inward from the exposed puncture hole adventitia.

**Table 2 Electron microscopy (EM) parameters for image capture and analysis.**

| Method | Raw image pixel size (XY) | Z Step size | Raw file size | Pre-binning for segmentation |
|---|---|---|---|---|
| SBF-SEM | | | | |
| Standard res | 100 nm | 200 nm | 1500−2500 images, 22−186 Gigabytes | 8 × 8 |
| High res | 20 nm | 20 μm | 20−30 images, 6−10 Gigabytes | NA |
| WA-TEM | | | | |
| Standard res | 3 nm | Single planes | 400−800 images montaged, 15−30 Gigabytes total | NA |

NA *not applicable*.

limited information on platelet organization within the interior of the thrombus. As shown by SBF-SEM (Fig. 2c–f and Supplementary Movie 2), a substantial intravascular portion of the thrombus was enclosed in 2D by column-defined vaults and the puncture hole was platelet-capped on the extravascular side. In electron micrographs, the intravascular surface was delineated by the endothelial cell layer on the vessel wall, as indicated to the lower right in Fig. 2c (binned 100 nm XY pixel size image, Table 2). In the example shown, the intravascularly open puncture hole contained many entrapped, darkly stained RBCs. As shown in Fig. 2d, these RBCs were frequently distorted in shape and often resembled polyhedral RBCs, an indicator of pressure within the capped thrombus [for review, see[24]]. Instead of a plugged hole, we observed a hole that was partially reduced in diameter by the accumulation of adventitial tether-platelets and capped extravascularly (Table 1, see also Supplementary Fig. 4). Intravascularly, the vaulted, platelet-aggregate defined spaces, were often open to the vessel lumen. Platelets towards the periphery of the platelet aggregates typically stained light gray, indicative of platelet degranulation. As shown at higher magnification (Fig. 2d, 20 nm XY raw pixel size), the lightly stained areas indeed consisted of degranulated platelets that were often nearly devoid of cytoplasm, ultrastructure expected of procoagulant platelets[25] (see, also Supplementary Fig. 3). The outer, intravascular surfaces of the thrombus were often sheathed by a layer of more loosely packed platelets (see asterisks in Fig. 2c and Supplementary Fig. 4, 20 nm pixel size imaging for more detailed examples).

As visualized in 3D with and without RBCs (colored red), and indicated in Fig. 2e vs. Fig. 2f, as well as in an end-on view, perpendicular to blood flow (Supplementary Movie 2), the

vaulted spaces extended nearly over the width of the puncture hole and areas of degranulated platelets extended "vertically" within the 3D structure consistent with the p-selectin staining shown in Fig. 2b. Renderings were color-coded to bring out platelet features (for details, see figure legend and Supplementary Fig. 3). Most of the platelets within the thrombus were found to be tightly packed as were those in the extravascular cap that seals the hole to stop further bleeding (Fig. 2 rendering, see also Supplementary Fig. 4). As indicated in Fig. 3 (a, raw data example; b/c, rendered) the intravascular "crown" was a relatively compact portion of the entire thrombus. In striking contrast, the extravascular portion of the thrombus spread hundreds of micrometers out from the puncture hole (Fig. 3c, d). All five rendered 5 min thrombi showed a similar structural pattern and similar extravascular capping including the p-selectin antibody, negative control in which infused fluorescent p-selectin antibody normally included as a tracer was deleted (mid hole cross-sections, Supplementary Fig. 2b–e, and Supplementary Movie 3, transparent rendering set included to show the full complexity of 5 min thrombus). Quantitatively, similar partial hole filling outcomes were found in all 5 min thrombi examples (Table 1) with amounts of RBCs trapped within the puncture hole or intravascular vaults depending on how open the thrombus crown was to the vessel lumen.

**Venous puncture wound thrombus growth is initiated by localized platelet aggregation with "pedestal" formation, 1 min post-puncture examples.** As shown by 2P fluorescence microscopy in Fig. 4a, the 1 min thrombus consisted of a platelet-rich structure (CD41 stain, red) that leaves an open hole. At its base, there was fibrin accumulation (blue color). As before, these

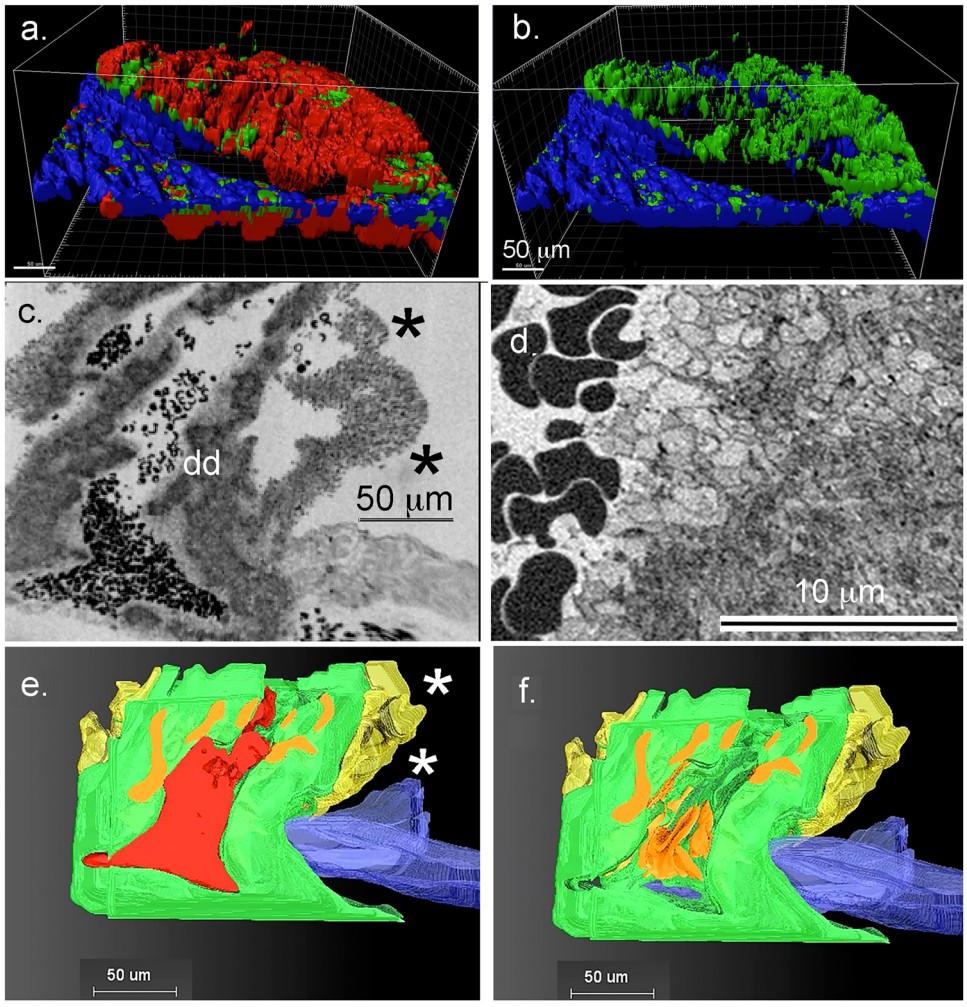

**Fig. 2 Bleeding cessation is through extravascular capping the vaulted 5 min post-puncture thrombus. a/b** Rendered images from 2P, light microscopy of a 5 min thrombus. **a** Three-color image: Red, CD 41 staining, a general platelet stain; Green, p-selectin antibody staining, an indicator of platelet α-granule secretion, Blue, fibrin. **b** Two-color: p-selectin (green) and fibrin (blue). Much of this staining is extravsascular[14], **c** Raw image, SBF-SEM, 100 nm *XY* pixel size in a raw captured image, mid-thrombus, showing an extravascular platelet layer that caps the puncture hole from the extravascular side. Asterisks, loosely adherent platelets. dd, approximate area shown in (**d**). **d** Raw image, SBF-SEM, 20 nm XY pixel size in a raw captured image of an area approximating dd in frame (**c**). Note that morphologically the platelets are most activated, i.e., degranulated on the surface of the column and least at the center. Furthermore, the trapped RBCs are distorted in shape, tending to polyhedral morphology. **e/f** 3D rendering of the thrombus with and without red blood cells (RBCs). RBCs, red; tightly adherent platelets, green, degranulated platelets, orange; loosely adherent platelets, yellow; vessel wall, blue.

results provide little to no detail at the cellular level and are entirely consistent with the previous work of Tomaiuolo et al.[14]. As shown in Fig. 4b, SBF-SEM provided considerably more detail on thrombus structure over length scales spanning hundreds of microns in the *XY* plane. The extravascular side of the punctured vessel wall of the jugular vein was marked by the accumulation of relatively short stubs, ~15 μm long, of platelet aggregates in the form of pedestals (P) with center-to-center spacings of roughly 20−30 μm. The height of the pedestals decreased with increasing distance from the puncture hole. We interpret the pedestals as sites of nucleated platelet−platelet accumulation with the gaps in platelet accumulation between the pedestals as evidence that platelet−platelet avidity is decidedly greater than platelet-adventitia avidity. Gaps between platelet aggregates/pedestals were also observed within the puncture hole (Fig. 5).

As platelet accumulation (Fig. 4) was tracked from the extravascular side into the puncture hole, a series of "bumps"

were observed suggestive of merged pedestals and outgrowth of platelet accumulation from a pedestal base. Within the hole (Fig. 4b', 2× magnification), areas of lightly stained strips of platelets (arrowheads) were observed, possible merger zones between enlarged pedestals. As the thrombus is then tracked into the intravascular lumen space, dark platelet aggregates project inward into the vessel in structures that are now 30−40 μm long, i.e., "columns" (C). Differences were observed in platelet staining intensity (Fig. 4b', see also Fig. 5 and Supplementary Fig. 2), suggestive of differences in platelet activation state, i.e., α-granule release. P-selectin antibody staining (Fig. 4a, green, intermixed, small puncta) indicated limited areas of α-granule release, totally consistent with previously published work[14]. We attribute the occurrence of electron dense RBCs trapped on the extravascular side of the thrombus to extravascular fibrin accumulation (present work and[14]). In total, we imaged four 1 min thrombi by SBF-SEM. As shown in Supplementary Fig. 2, each of the three

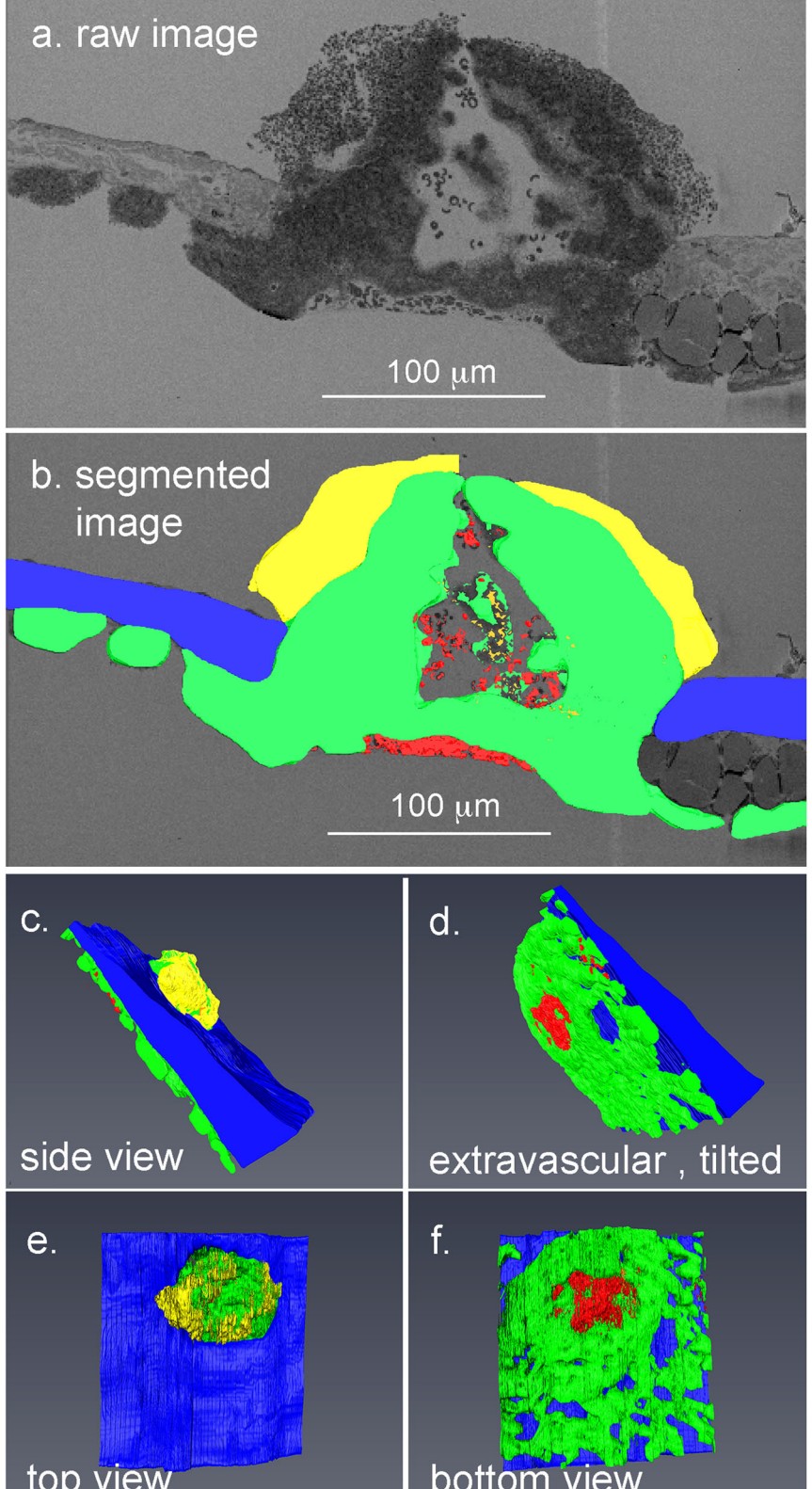

**Fig. 3 Overall 3D structure reveals limited loosely adherent platelet sheathing of 5 min post-puncture wound thrombi. a/b** Example of single segmented image slice from a 5 min post-puncture jugular vein thrombus showing loosely adherent platelets (yellow), tightly adherent platelets (green), vessel wall (blue), RBCs (red) trapped intravascularly with a vault, and a small RBC patch on the extravascular presumably entrapped by extravascular fibrin[14]. Flow from left to right. **c–f** Full 3D rendering showing various features of the thrombus when viewed from different angles. Loosely adherent platelet sheathing of the intravascular thrombus crown is limited to small patches. RBC (red), tightly adherent platelets (green), degranulated platelets (orange), loosely adherent platelets (yellow), vessel wall (blue).

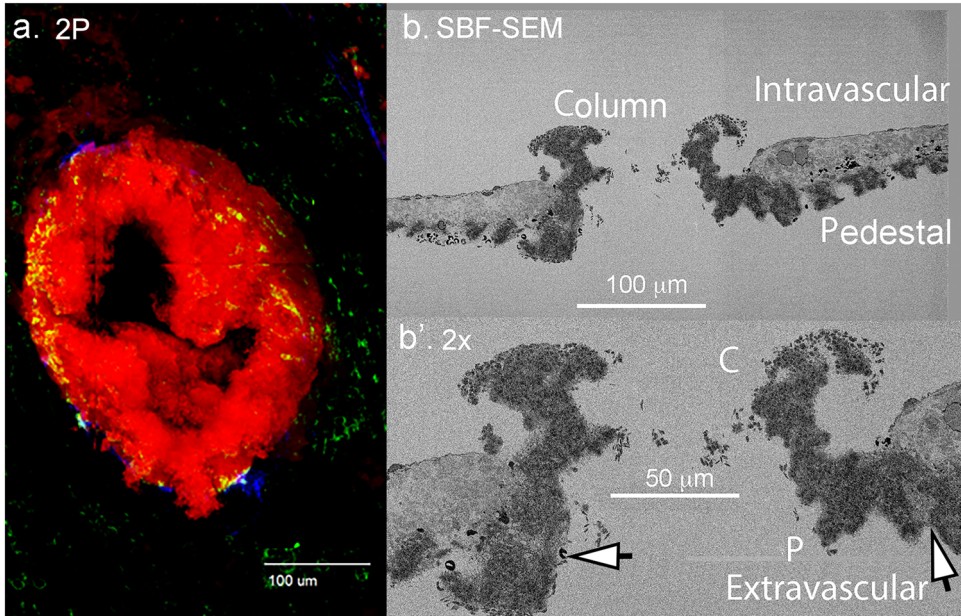

**Fig. 4 Early, nucleated platelet accumulation as revealed by 2P and SBF-SEM imaging of 1 min post-puncture thrombi. a** 2P, light microscope imaging of the thrombus, maximum intensity projection. Red, CD41 antibody staining, a marker for platelet accumulation; green: p-selectin antibody staining, a marker for α-granule fusion with the platelet plasma membrane; blue, fibrin antibody staining, a marker for what is principally extravascular accumulation of fibrin (see[14]). **b/b'** SBF-SEM visualization, 20 nm raw *XY* pixel size, one of a series of images taken every 20 μm across the forming thrombus. The intravascular lumen of the mouse jugular vein is to the top and the extravascular side of the vessel wall is to the bottom. The vein is lined by a layer of endothelial cells and collagen-rich adventitia below in these images. C, columnar platelet aggregate; P, pedestal (apparent nucleated platelet accumulation anchored to vessel wall adventitial layer. Arrows point to lightly staining platelet areas indicative of platelet degranulation. These presumably correspond to mobilized/ exposed p-selectin staining in the 2P image. Electron micrograph slice is parallel to flow.

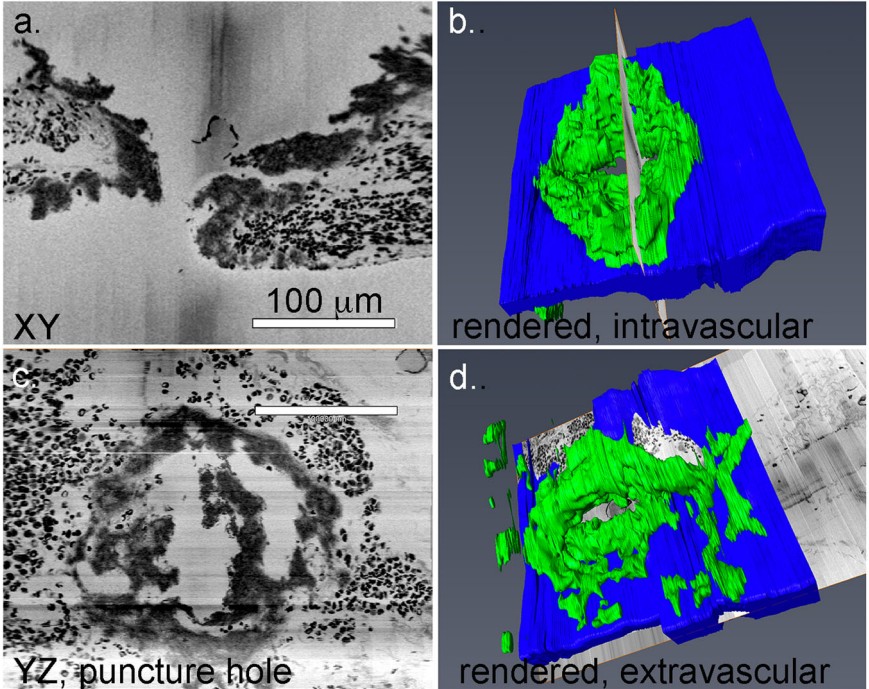

**Fig. 5 Morphological evidence for pedestal extension as the starting point for intravascular growth and puncture wound hole sealing.** *XY* cross-section slice through a 1 min post puncture wound hole (**a**) and its placement within the thrombus (**b**). *YZ* cross-section within the same puncture hole seen as imaged (**c**) and placed within a rendering (**d**). Raw images are 100 nm pixel size, 200 nm spacing in *Z*-dimension. Viewed from either orientation, the extension of the pedestals through further platelet aggregation appears to be a starting point for further thrombus growth. See also Supplementary Movies 1 and 4.

additional examples displayed similar overall structural features including the negative antibody control in which p-selectin antibody, potentially an inhibitory antibody, was deleted during the preparative procedure (Supplementary Fig. 2c).

To provide a clearer 3D structural view of early thrombus formation after the venous puncture, we viewed individual SBF-SEM slice features within the context of 3D thrombus reconstructions. As shown in Fig. 5a, pedestals and columns were obvious in the puncture wound hole in a second example viewed in cross-section and placed within a 3D context (Fig. 5b, view from vessel lumen, green tightly packed platelets, and blue vessel wall). To peer through the hole, we tilted the display of the raw data to provide a *YZ* view of the surface and hole. As shown in Fig. 5c, d, platelet pedestals projected inward and partially filled the hole leaving gaps about the rim of the hole (Fig. 5c, Supplementary Movie 4, and Supplementary Fig. 2b). For a full 3D rendering of the thrombus hole in which sequential images of the raw SBF-SEM within the hole are shown, see Supplementary Movie 4. In conclusion, overall, 1 min thrombus formation was highly reproducible.

**Puncture wound thrombus formation and remodeling show sensitivity to both P2Y$_{12}$ and thrombin inhibitors**. The 5 min post-puncture experiments leave an unanswered question as to whether the puncture wound thrombi remain vaulted over longer periods of time. To investigate this question, we characterized thrombus structure at 20 min after jugular vein puncture in four animals. As shown in Fig. 6a, b, the wide-area TEM images from two of these samples, one parallel and one perpendicular to the blood flow, and two SBF-SEM datasets from full thrombi (Fig. 6c) showed the presence of vaulted regions, albeit with a reduced

number in the 20 min thrombi. As suggested by the tight central packing in the wide-area TEM images, the volume of the vaults may be restricted by compression. Consistent with the suggestion of possible compression of the thrombus interior, we found by 2P immunofluorescence that fibrin within the interior of the clot was concentrated towards the central p-selectin positive zones within the thrombus (Fig. 6d). Overall, the 20 min thrombus showed features indicative of considerable thrombus remodeling with time. Additionally, in the compressed central areas evident from WA-TEM images as well as SBF-SEM images (Fig. 6), the crown of the intravascular thrombus was sheathed by multiple layers of loosely adherent platelets.

The recruitment of loosely packed platelets to the crowning surface of the forming thrombus was apparent at 5 min post-puncture and revealed as an intravascular sheath of loosely packed platelets at 20 min post-puncture. This loosely adherent platelet sheath, at least superficially resembles morphologically, the ADP- and thromboxane-dependent platelet shell morphology of the Core and Shell paradigm in which formation of the Core is sensitive to thrombin inhibition while later-formed Shell is selectively sensitive to P2Y$_{12}$ inhibition. To test whether wound thrombus formation occurs by similar signaling mechanism(s), we took two parallel approaches. We pre-treated mice with either the direct-acting P2Y$_{12}$ inhibitor, cangrelor[14], or with the direct-acting anti-coagulant (DOAC), dabigatran, a thrombin inhibitor[20], followed by subsequent puncture wounding with a 30-gauge needle in the presence of the drug. We found that cangrelor strongly inhibited bleeding cessation and most steps in thrombus formation; thrombus formation did not extend past formation of a circumferential ring of recruited platelets lining the puncture hole (osmium black staining, Fig. 7a). Our results are consistent with those observed by Tomauiolo et al.[14] in which

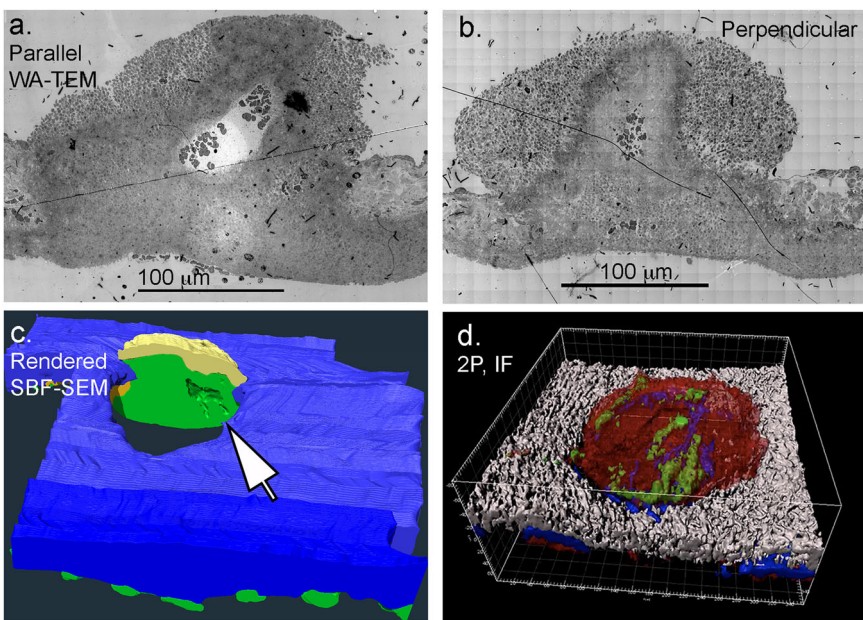

**Fig. 6 Thrombus remodeling leads to the retention of vaulting in 20 min thrombi.** The central portion of the thrombi appears to contain a "compressed" zone of tightly adherent platelets. **a/b** wide area TEM images of 20 min post wounding jugular thrombi taken either parallel to flow or perpendicular to flow. Note that the parallel image appears shaped by flow while the perpendicular to flow image appears to be nearly symmetric with respect to left and right. **c** A split open 3D rendering from SBF-SEM imaging of a 20 min post puncture thrombus (representative example, n = 2, SBF-SEM image sets) showing that the limited vaulting in the thrombus is enclosed by tightly adherent platelets. Tightly adherent platelets (green), loosely adherent platelets (yellow), vessel wall (blue). Note that extravascular platelet accumulation is found in hundreds of microns from the ~200 μm diameter puncture hole. **d** 2P, immunofluorescence rendering of protein distributions within a representative 20 min thrombus. Platelets (red, CD41), mobilized/exposed p-selectin (green), fibrin (blue), vessel wall (white). Note that fibrin staining "bands" within the intravascular interior of the thrombus appear to be in proximity to parallel bands of mobilized/exposed p-selectin staining. WA-TEM, wide area transmission electron microscopy. SBF-SEM, serial block face scanning electron microscopy. If, immunofluorescence.

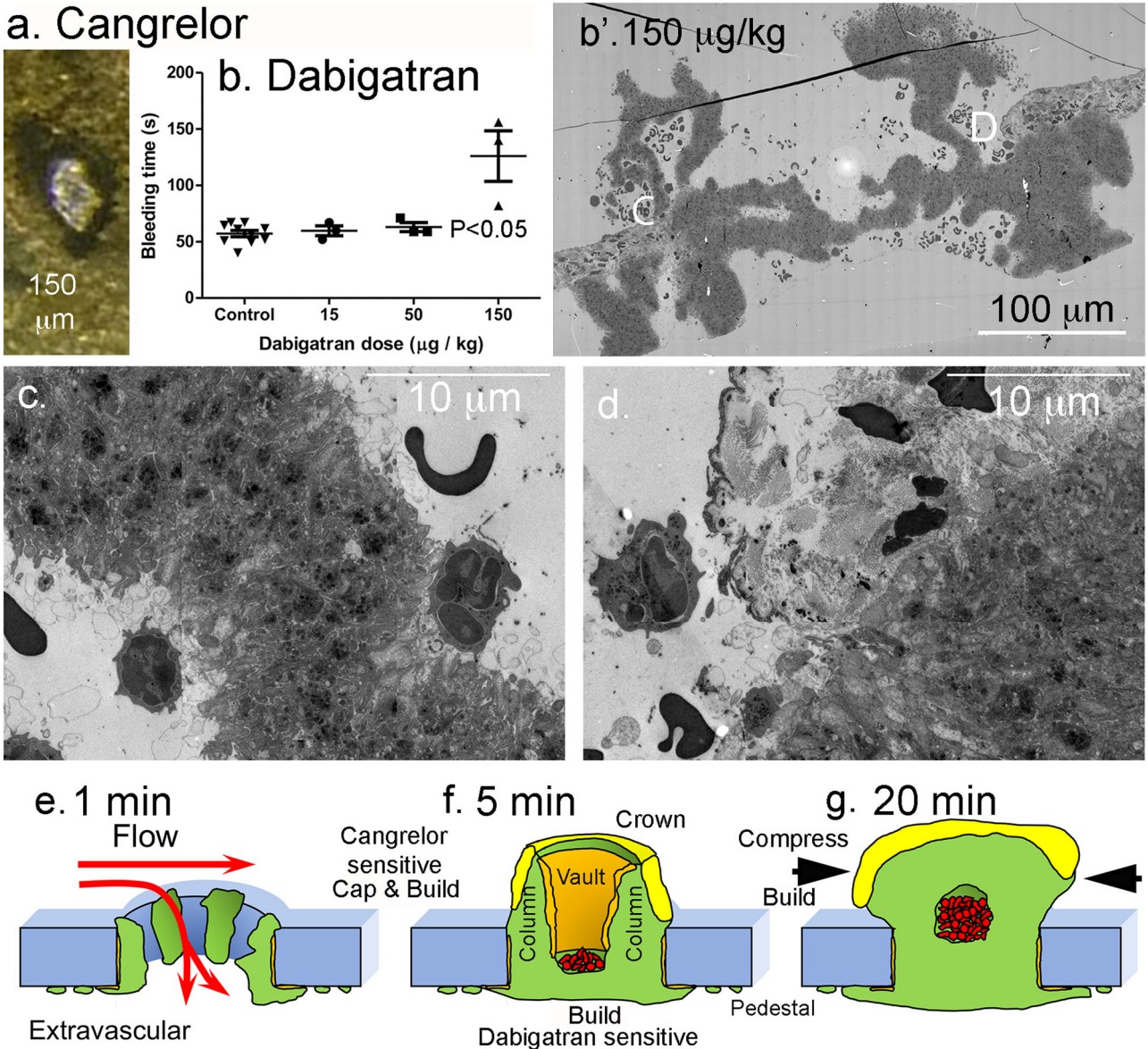

**Fig. 7 Drug sensitivity of thrombus formation. a** Cangrelor treatment has a major effect on 5 min thrombus formation with osmium black platelets accumulating only around the rim of the puncture hole. **b** Dabigatran dose-response bleeding curve. Mean plus or minus standard deviation. **b'** WA-TEM (wide-area transmission microscope) cross-section across the full puncture, dabigatran at 150 μg/Kg. WA-TEM, 3 nm raw pixel size. **c/d** Blowups of areas noted with C and D in (**b'**). Light staining, degranulated platelets are at most a very thin layer on the column surfaces with dabigatran treatment. Note that apparent PMN (polymorphonucleocyte) attachment, although rare, is more obvious in dabigatran treated cells with PMNs appearing to make attachments to degranulated, procoagulant platelets through pseudopods. **e–g** A working model, Cap and Build, green: tightly adherent platelets, orange: areas of degranulated platelets obvious by SBF-SEM, yellow: loosely adherent platelets., red: trapped RBCs, RBCs are seen in variable amounts in 5 min thrombi as all preparations are in situ saline washed and intravascular continuities to vessel lumen are variable in amount.

most cangrelor treated 30 gauge needle examples continued to bleed at 5 min post-puncture. Both sets of outcomes suggest that P2Y$_{12}$ activity is essential during multiple steps in thrombus formation leading to thrombus crowning and the recruitment of loosely adherent platelets. We speculate that at these drug levels, effects are likely due to a lack of integrin-dependent platelet adhesion as indicated by the lack of column and vault formation. Interestingly, we found at a moderate dabigatran dose, sufficient to cause a mild, but statistically significant inhibition in bleeding cessation time (Fig. 7b), important variations in thrombus formation, 5 min post-puncture. Intravascular column formation was distorted with the columns appearing "wobbly" with platelet

degranulation limited to the surface layer of column lining platelets and little recruitment of loosely adherent platelets (Fig. 7b'−d). On the other hand, polymorphonucleocyte (PMN) recruitment appeared to be enhanced as was extravascular column formation. Importantly, tight platelet−platelet adhesion within the central interior portions of columns was little affected suggesting active integrin-dependent platelet adhesion. In brief, our data point to direct or indirect thrombin-dependent processes playing an important role in platelet recruitment and the sheathing of the extravascular crown in loosely adherent platelets. This outcome is not expected in a Core and Shell paradigm, and therefore suggests significant signaling aspects of what we term a

Cap and Build paradigm (Fig. 7e–g) for puncture wound thrombus formation.

## Discussion

These experiments were initiated with the perhaps naïve assumption that platelet recruitment to seal a puncture wound might be governed by a different set of geometric and structural principles than those of other forms of vascular damage. In the puncture wound, amongst the oldest form of vascular damage known to humans, the vessel wall is penetrated leaving exposed an adventitial, collagen-rich matrix lining the puncture hole. Platelets are then recruited upon this surface and the forming platelet-rich thrombus seals the hole leading to bleeding cessation. The currently dominant Core and Shell model of platelet response to vascular damage predicts that the hole becomes sealed by a cylindrical infill resulting from progressive adhesion of platelets to one another, building inward from a rapidly formed, collagen-bound, and uniformly distributed platelet layer lining the hole. In brief, the expected outcome is the formation literally of a platelet plug. Taking a correlative light and 3D electron microscopy approach combined with WA-TEM to define full 3D thrombus structure at electron microscope resolution (here varying from 100 to 3 nm $XY$ pixel size) and map the distribution of important hemostatic proteins, we found that primary hemostasis in response to a puncture wound, 300 μm nominal diameter, mouse jugular vein, was structured about a hitherto unrealized vaulted, platelet-rich thrombus in which a platelet cap sealed the hole from the outside of the vessel without actually filling the hole. In brief, a vaulted thrombus does not form a plug. Overall, our findings suggest a hitherto unknown set of structural principles for a puncture wound thrombus formation in which platelet aggregates, be they, initially forming pedestals or longer columns, generate a vaulted wound thrombus. Here, we discuss the implications of these results, together with a proposed paradigm to explain them, and possible implications of our results for anti-platelet therapies and local versus systemic drug administration.

At 1 min post puncture, all structurally examined jugular vein thrombi were open on both their intravascular and extravascular sides, i.e., the puncture hole was neither filled nor capped. By 5 min post-puncture, the bleeding had stopped in all cases. Importantly, when examined spatially over the 1500−2500 SBF-SEM image slices used for full 3D rendering of the thrombi, the 1 min thrombus consisted of collagen-anchored platelet aggregates, pedestals, spread at roughly uniform spacing across the extravascular adventitial surface. Within the wound hole, larger aggregates were typically pressed one against another with bands of apparently degranulated platelets appearing almost like mortar between the aggregates. Even within the wound hole, pedestals were often found separated by platelet-free adventitia. We refer to the larger aggregates found on the intravascular side of the hole as columns. These results suggest that platelet aggregation is nucleated and that the avidity of platelet−platelet binding appears to be greater than platelet-adventitia, i.e., collagen binding, be it direct or indirect. In brief, early thrombus formation is structured about pedestals and columns, which appear morphologically to be the site of subsequent platelet recruitment. These pedestals and columns form the capped structure from which the 5 min thrombus is built. We note that these features were not resolvable in our, parallel two-photon light microscope images, nor can they be visualized by conventional scanning electron microscopy in which only the thrombus surface is visualized. The observation that platelet degranulation is confined to a small subset of morphologically-identified, lightly staining platelets is supported by the patchy, very limited distribution of p-selectin staining seen by immunofluorescence.

We predicted that pedestal and column formation would lead to a 5 min post-puncture thrombus structured about a vaulted

interior of partially merged pedestals and extended columns and an extravascular platelet cap that sealed the wound hole from the outside. That outcome was indeed observed for all five SBF-SEM datasets. Each was sealed from the outside by a platelet-rich extravascular cap, while a central cavity within the puncture hole was devoid of platelets and continuous with the vaulted intravascular vessel lumen. Continuity with the vessel lumen was demonstrated in two different ways: (1) by full ultrastructural analysis and (2) from the observation of variable quantities of RBCs being trapped on the intravascular side of the hole. In these washed preparations, the extent of RBC accumulation was inversely correlated with the extensiveness of continuities with the vessel lumen. Extensive intravascular vaulting was apparent within the intravascular thrombus crown. A similar structural organization was observed in the presence or absence of infused p-selectin antibody infusion. Zones of platelet degranulation were most pronounced along column peripheries and apparent merger zones between platelet aggregates. This appearance was consistent with the 2P immunofluorescence imaging. Discrete, anchored platelet pedestals were found in the more distal portions of the extravascular thrombus structure. In sum, our EM results and 2P p-selectin immunofluorescence pointed to extravascular capping of the puncture hole as a central process in bleeding cessation, which implies a novel set of structural principles upon which a puncture wound thrombus is built.

The organization of the thrombus at the final time point, the 20 min puncture thrombus, raises three intriguing points. First, intravascular vaulting extending into the puncture hole was still apparent. This tended to be centrally located within what looked like a central, compressed pier of platelets that was most obvious in WA-TEM images collected perpendicular to flow. This raises the question of whether reduced vaulting is due to infill or structural compression. Fibrin accumulation and clot retraction present a possible mechanism. Second, on the intravascular side, the compact central thrombus area was sheathed in a multi-plate thick layer of loosely adherent platelets that extended in limited areas deeply into the interior of the thrombus. These findings are suggestive of pronounced differences in signaling following extravascular thrombus capping. Third, extravascular accumulation of platelets must be an ongoing process; large-scale extravascular accumulation of platelets was readily visualized in the images. One overall explanation of these observations is that the thrombus itself is not a rigid structure but one that is deformable and, more speculatively, a dynamic structure in which platelet movement/migration from intravascular to extravascular side might be occurring. We note that neither our EM data nor our correlative light microscopy [see also[14]] provides any evidence for a highly activated platelet "core" that fills the puncture hole as predicted by a Core and Shell paradigm[16]. Consistent with this distinction, our drug experiments point to $P2Y_{12}$ signaling being important early in thrombus formation with the directly acting inhibitor, cangrelor, stalling thrombus formation at what appears to be a tethering/docking stage of platelet adherence to the exposed vessel hole adventitia and the DOAC, dabigatran, a thrombin inhibitor, affecting column formation and recruitment of loosely adherent platelets, early steps in sheath formation. The full implications of these results and their potential translational importance will require further detailed experimentation.

In sum, our evidence from correlative light and 3D electron microscopy together with supporting WA-TEM reveal a coarse-grained structure sufficient to support three major findings. First, our data show that bleeding cessation following venous puncture is due to extravascular capping of the puncture hole, rather than infill. Indeed, our mothers were right that the scab is important. Second, our venous studies point to principles of thrombus formation, i.e., pedestal and column formation leading to a vaulted

thrombus, which we conceptualize in a Cap and Build paradigm in Fig. 7. Third, the results here are also likely applicable to traumatic injury and suggest that enhanced bleeding cessation might be best achieved by local, external therapeutic application. In brief, the sterile gauze applied to the wound could well become the vehicle for localized drug delivery. Our proposed mechanisms for bleeding cessation in venous puncture might be generalizable to other systems, such as injury of arteries or arterioles and thrombosis. In support of this contention, we point to the recent Communication from the ISTH SSC Subcommittee on biorheology bringing attention to "the fact that that the relevant range of average wall shear rates in human arteries where clinically relevant arterial thrombosis occurs may fall as low [as] significantly overlapping with what are considered "venous" shear rates"[26]. Furthermore, as illustrated in Supplementary Fig. 5, our structural approaches and outcomes can be explicitly tested in arteries, even under conditions in which rebleeding leads to fragmentation of the extravascular thrombus organization. One could well speculate that structural work may have clinically relevant carryover and serve as in vivo gold standard against which to reference reductionist approaches such as microfluidic models of thrombus formation. The effects of cangrelor and dabigatran on thrombus formation, the first acting early and preventing capping and intravascular thrombus crowning, i.e., both cap and build, and the second in strongly affecting the thrombus build and as a corollary enhancing PMN recruitment, may well indicate unanticipated drug side effects that deserve further consideration.

In closing, we wish to emphasize that what is presented here is a first step in the structural understanding of bleeding cessation. The data lead to what in structural terms would be considered a "coarse grain" structural model. The level of resolution of our 3D structural renderings, albeit informed by more detailed 3 or 20 nm imaging, is that of the individual platelet rather than its organelles or molecules. In molecular terms, this might be considered to be the equivalent of a model of the protein $\alpha$−helices rather than an atomic model. In the future, though, increasingly detailed levels of structural definition will be achieved through approaches such as higher resolution 3D imaging of regions of interest to define, e.g., the in situ shapes that platelets assume and with emerging instrumentation full thrombus imaging at an even more detailed level. What we chose to accomplish was what was practical with existing resolution and computational power. For rendering, we chose to reduce date sets to at most a 1 or 2-gigabyte size to make near manual segmentation through image stacks of 1500−2500 slices practical. We chose to concentrate on one defined and reproducible example that of mouse jugular vein puncture formed by a 30-gauge needle. In the future, we will be able to answer what happens when the wound hole is small and when the hole is too large to be capped, and numerous other questions. For now, we freely acknowledge that an "atomic model" of a 5 min thrombus at $5 \times 5 \times 5$ nm, a ~1000 terabyte dataset, is conceivable but presently impractical.

## Methods

**Mice and reagents**. All animal usage protocols were approved by the respective Institutional Animal Care and Use Committee (commonly termed IACUC) at the University of Pennsylvania or the University of Arkansas for Medical Sciences. 8−12 week old, wild-type male or female C57BL/6 mice were used in equal numbers across the individual experimental time sets. All reagents have been cited[14,27,28].

**Drug treatments**. Cangrelor and dabigatran were purchased from Sigma-Aldrich (St. Louis, MO), and prepared as 5 mg/mL stock solutions in water and 0.1 N HCl, respectively. The final working solutions were diluted in normal saline. Cangrelor was infused as an initial bolus dose of 0.3 mg/kg followed by 0.1 mg/kg/min maintenance with a syringe pump via a jugular venous catheter[14]. Dabigatran was

injected into the jugular vein at a bolus dose starting at 15 μg/kg[29] and increasing to 150 μg/kg, 20 min before puncture.

**Thrombus preparation**. Antibodies for correlative light and electron microscopy (anti- fibrin, CD41, and p-selectin) were perfused prior to jugular vein puncture and all preparative procedures were as previously described[14]. The wound was with a 30-gauge needle, 300 μm diameter nominal wound size, and thrombi were fixed in situ at 1, 5, and 20 min with 4% paraformaldehyde.

For SBF-SEM, samples, initially fixed with 4% PFA, were pinned to silicon mats and fixed in 0.1 M cacodylate buffer containing 2.5% glutaraldehyde and 2 mM calcium chloride for 1 h in ice. Thrombi were heavy metal stained[27,30]. In brief, after washing three times with cold 0.1 M sodium cacodylate buffer containing 2 mM calcium chloride samples were fixed in reduced osmium containing 3% potassium ferrocyanide in 0.2 M cacodylate buffer with 4 mM calcium chloride with an equal volume of 4% aqueous tetroxide for 1 h in ice. Samples were then placed in a 0.22 μm-Millipore-filtered 1% thiocarbohydrazide (TCH) solution in double-distilled water for 20 min following five washes with double-distilled water at room temperature each for 3 min. Samples were postfixed in 2% osmium tetroxide in double-distilled water for 30 min at room temperature following five washes with double-distilled water at room temperature each for 3 min. Samples were then placed in 1% uranyl acetate (aqueous) and left overnight at 4 °C. The next day, samples were washed five times with double-distilled water at room temperature each for 3 min and processed for en bloc Walton's lead aspartate staining. Samples were placed in lead aspartate solution and were transferred to a 60 °C oven for 30 min, following five washes with double-distilled water at room temperature each for 3 min. Samples were then dehydrated and processed for resin embedding.

### Correlative 2P and SBF-SEM microscopy

*2P light microscopy*. Antibody and 2P light microscopy procedures have been cited[6,14]. Samples were 2P imaged before processing for SBF-SEM electron microscopy.

*Antibody concentrations*. Antibodies were infused prior to jugular puncture: anti-CD41 F(ab)2 fragments (0.12 μg/g body weight; clone MWReg30, BD Bioscience), anti-P-selectin (0.2 μg/g body weight; clone RB40.34, BD Bioscience), and anti-fibrin (0.2 μg/g body weight; clone 59D8)[6].

**SBF-SEM and image segmentation/analysis**. The stained 300 μm diameter puncture wound with its surrounding tissue occupies a volume of approximately 1 mm³, which can be seen easily in the amber-colored epon block. The specimen of thrombus was cut out of the epon block and trimmed for mounting onto an aluminum pin, such that the direction of the blood flow was perpendicular to the motion of the ultramicrotome knife in the SBF-SEM. Imaging was performed with a Gatan 3View system mounted inside a Zeiss Sigma SEM[23,24]. To acquire image planes through the entire puncture wound (300−500 μm depth) data were collected with a 100 nm pixel size in XY and 200 nm Z-step. To provide an ultrastructural reference for the blood clot, a higher resolution image was collected with a 20 nm pixel size in XY, for every 20 μm in Z (100 slices). In this way, a total of 1500−2500 sections were obtained. Each of the 15−25 data sets were aligned separately and then combined and aligned with each other. Some of the data were obtained by using FCC (focal charge compensator) to minimize electrical charging in the specimen. The data were then binned and imported to the Amira software (version 2020.2, ThermoFisher FEI) for segmentation and again aligned relative to the puncture hole. The imaged clot was segmented according to the following: vessel wall (blue), tightly adherent platelets (green), loosely adherent platelets (yellow), degranulated platelets (orange), and RBCs (red). Segmentation and data analysis were performed as described[27,28,31]. Volumes were computed from summating voxels within the segmented objects[27,28].

**Wide area-TEM**. individual thrombi, initially fixed with 4% PFA, were pinned to the silicon mats and processed with minor modifications[32]. In brief, samples were fixed for 20 min on ice with 2.5% glutaraldehyde and 0.05% malachite green (EMS) in 0.1 M sodium cacodylate buffer, pH 6.8. Samples were post-fixed for 30 min at room temperature with 0.5% osmium tetroxide and 0.8% potassium ferricyanide in 0.1 M sodium cacodylate, washed, stained for 20 min on ice in 1% tannic acid, washed with H₂O, and incubated and for 1 h in 1% uranyl acetate at room temperature. Specimens were dehydrated with a graded ethanol series, and embedded in Araldite 502/Embed 812 resin (EMS). Each plug was cut at 10, 25, and 50% depth sequentially, ultrathin sections were placed on slotted grids coated with formvar film (EMS) and imaged at 80 kV on FEI Technai G2 TF20 transmission electron microscope. Digital images were acquired with FEI Eagle 4k CCD camera. Montages images were obtained using SerialEM software (version 3.6, 32 bit, University of Colorado, Boulder, CO) and assembled with IMOD software (version 4.9.13, University of Colorado, Boulder, CO).

**Statistics and reproducibility**. Data were analyzed using Student t-tests, or when appropriate, using one-way ANOVA *post hoc* Bonferroni *t*-tests implemented in

GraphPad Prism 5 (San Diego, CA). All experiments were reproduced by imaging at least four independent thrombi for time interval tested. SBF-SEM and WA-TEM data product independent ultrastructural approaches for the data at each time interval.

**Reporting summary**. Further information on research design is available in the Nature Research Reporting Summary linked to this article.

## Data availability

Wide area transmission electron microscopy (WA-TEM) at 3 nm *XY* resolution and 3D SBF-SEM image series at 20 and 100 nm *XY* have been deposited in the Electron Microscopy Public Image Archive (EMPLAR ID (accession code) 10785). The source data for Figs. 1 and 7 dot plots can be found in Supplementary Data files. All other raw data are available by request from the corresponding author.

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

## Acknowledgements

Thanks are given to all team members for what was truly a team effort over a number of years. Initial puncture wounds and supporting light microscopy were done by Dr. Tim Stalker at the University of Pennsylvania. We would like to thank Dr. Sidney W. Whiteheart for his comments during the course of this work and the preparation of this manuscript. We thank Lawrence F. Brass for this suggestions on the wording of the title. Work at UAMS within the Storrie and Rhee laboratories was supported by NIH grants R01 HL119393, R56 HL119393, and R01 HL 155519 to BS and P01 HL146373 (Lawrence F. Brass, PI, University of Pennsylvania, subaward to BS). The Leapman laboratory was supported by the intramural program of NIBIB at the National Institutes of Health, Bethesda, MD. Work at the University of Pennsylvania was supported by NHI grants P01 HL040387 and P01 HL120846 to TJS and LFB.

## Author contributions

S.R. and I.P. contributed equally to experimental implementation with major roles in data gathering. K.B. was responsible for puncture wounds and all animal surgeries at UAMS. All SBF-SEM was done by the Leapman laboratory at NIH, NIBIB with K.L., Y.V., J.C., D.C. and O.Z. responsible for experimental implementation and M.A.A. and R.L. being major contributors to experimental design and supervision. All electron microscope sample preparation and WA-TEM were done by I.P. who was also responsible for data validation and coordination between the Storrie and Leapman laboratories. J.A.K. and G.Z. supported sample loading and electron microscope operations. R.L., S.R. and B.S. had major responsibilities for experimental design, data quality control, and paper preparation. Brian Storrie prepared the first draft and took senior author responsibility for seeing the paper through to completion.

## Competing interests

The authors declare no competing interests.
