## [Transparent Peer Review File · Communications Biology]

Reviewers' comments:

Reviewer #1 (Remarks to the Author):

The manuscript by Rhee and colleagues is focused on an hemostatic response after a needle puncture of the jugular vein of adult mice. The authors describe a platelet plug morphology that was named "vaulted thrombus" which presents an inverted "U" shape and which is capped by platelets. The authors propose a new "Cap and Build paradigm that should have translational implications for bleeding control and hemostasis". While the authors used very powerful experimental approaches notably high quality imaging, and describe an unexpected and interesting platelet plug morphology, some results provided are not novel and quite expected, notably the sensitivity to anti-thrombotic agents or the presence and localization of pro-coagulant platelets. Moreover, the fact that they used only one approach to injure a vessel and only one vessel type is a major drawback of this study and the plug morphology described here cannot only based on this description represent a new paradigm.

Major concerns:

1. The thrombus formation during an hemostatic response after penetrating injury has already been studied in the femoral artery (Welsh et al., 2016) and was described to present a "Core and Shell" structure. It seems essential that the authors study the hemostatic thrombus formation in other vessels after penetrating injuries to demonstrate the universality of a "Cap and Build" paradigm.
2. Can the authors explain why they proposed a new paradigm of "cap and build" while using the exact same model Tomaiuolo et al (2019) described a core and shell model? Is it not because the authors observed earlier time points compared to the other study (Tomaiuolo et al., 2019)? This would mean that there is no novel paradigm, but just an initial structure which does not exactly correspond to the core/shell model, but becomes a core/shell at late stages? We also wonder if the steps of thrombus formation are distinct for small (35-gauge) and big (23-gauge) injuries (Tomaiuolo et al., 2019)?
3. It seems that the authors believe that the vault-lining of degranulated platelets are procoagulant platelets. This is not strictly demonstrated and a better image quality would help. Could the authors use AnV or other markers for procoagulant platelets?
4. Almost every experimental thrombosis model is dependent on anticoagulants, it is therefore not surprising that thrombosis in this model relies on dabigatran, and the link with the presence of PP in the vaulted region and the effect of an anticoagulant is far from being obvious.
5. The authors propose a new paradigm, but it seems that in the plug they describe that most activated platelets are in the center (core) and the loosely attached less activated platelets are on the edge (shell), which is not that different from the core and shell paradigm.
6. It would have been very informative to obtain some spatio-temporal information using intravital microscopy. The so-called pedestals seem to form at first, which is not very surprising as they expose the matrix where platelets adhere to.
7. Figure 2: the quality of the EM images are very poor. They are probably good enough to localize RBC, but not informative enough to provide insight on platelets (level of activation, degranulation...). Fig 4B: darker staining is not a formal proof of stronger activation. Can the authors not focus and look at the granule content as a sign of activation level ?
8. Movies were not provided.
9. Can the authors explain why they observe bleeding with cangrelor while Tomaiuolo et al., 2019, had a much minor effect? According to text, in the current manuscript almost no platelet plug was detected, while the plug was clearly shown in the study of Tomaiuolo et al., 2019.
10. The authors wrote: "the "Cap and Build" paradigm could affect on the existing treatment protocols because it indicates that enhanced bleeding cessation might be best achieved by local, external therapeutic application". This comment is surprising as covering the hole with sterile gauze piece is a standard procedure and not novel at all.
11. Based on images in Fig 3 and supplement Fig 2, it seems that the hole could actually be filled. Again providing images over time or videos appears critical if the authors want to prove that in their new paradigm the first step of the process is not "filling the hole".

Minor comments:

1. Methods: what is the proof that the lesion exposes collagen and TF as proposed by the authors?

2. Methods: n=4 is a very low number of experiment for experimental thrombosis with the objective to determine injury size and detail the morphology of the plug? I would recommend to increase the number of experiments.
3. Figure 1: it is not easy to understand what we look at in Figure 1D
4. Out of curiosity, the authors called the plug "thrombus": could the authors explain why they did not use the term plug as they observe an hemostatic process, i.e. a biological response after injury of a healthy vessel wall?
5. The fact that a thrombus rich in platelets contains region with RBC that are squeezed is not really surprising.
6. The authors describe a Swiss cheese like interior with holes, but do they do not provide an idea of the number of holes per thrombus?
7. The authors wrote that the platelet aggregate formed after vessel rupture does not form a "plug". A plug means connecting and if the thrombus formed is not similar to the one occurring after non-penetrating injuries, the aggregate is still sealing the breach by forming a connected structure, which could be considered as a plug.

Reviewer #2 (Remarks to the Author):

Comments:

There are three major findings in the present study: 1) Bleeding cessation following venous puncture is due to extravascular capping of the puncture hole, rather than filling the hole to form a solid platelet plug; 2) the authors propose a novel principle of thrombus formation, that consists of pedestals and columns of vaulted structure thrombus with a central cavity. Thus, the suggested architecture of the thrombus structure in the present study dismisses the earlier concept of core and shell principle of thrombus formation; 3) the early steps in thrombus formation are sensitive to P2Y₁₂ inhibition by cangrelor, and late steps to thrombin inhibition by dabigatran. The evidence in support of the Cap and Build principle of thrombus formation comes from the studies that employed correlative light and 3D electron microscopy together with supporting WA-TEM. The results of the study strongly support the Cap and Build hypothesis of thrombus formation following venous puncture. However, the authors may address the following queries.

1. Referring to the Introduction section, the statement "while extravascular thrombus capping was sensitive to the directly acting P2Y₁₂ ADP-receptor inhibitor, cangrelor" is not in accordance with the label of Fig 7F (Cap Depigatran-sensitive). This may be looked into.
2. Legend to the Table: "Cavity dimensions decrease with time". If one compares the central cavity diameter between 1 and 5 min post-puncture, the values shown are not statistically significant. Then, how is it that cavity dimensions decrease with time?
3. The diameter of the central cavity of the vaulted structure is given in the Table. However, it is of geometrical interest to determine the height of columns and pedestals in order to understand the stability of the thrombus structure.
4. "The height of the pedestals decreased with increasing distance from the puncture hole". How was the height of pedestals determined? A graph illustrating the height of pedestals vs distance from the puncture hole will provide a quantitative expression of the results depicted in Fig 4B.
5. In Fig 7B, the data with respect to the dose-related effect of dabigatran on bleeding time are shown. The authors should also provide the data relating to the effect of cangrelor on the bleeding time.
6. Legend to Fig 7G: The schematic diagram shown in Fig 7G is not described in the figure legend.
7. It is described in the results section that "Intravascularly, the vaulted, platelet-aggregate defined spaces, were often open to the vessel lumen". What will be the impact of the blood flow (shear stress) inside the vessel on the thrombus structure?
8. Discussion section: "while a central cavity within the puncture hole was devoid of platelets and continuous with the vaulted intravascular vessel lumen." "more speculatively, a dynamic structure in which platelet movement/migration from intravascular to extravascular side might be occurring". There appears to be some discrepancies in the observed result and speculation about the platelet movement in the dynamic structure.
9. Will the size of injury influence the architecture of thrombus structure and the Cap and Build model of thrombus formation, suggested in this investigation?

Reviewer #3 (Remarks to the Author):

This is an elegant descriptive study of bleeding and hemostatic plug formation at the site of a puncture wound. This adds to the now established 'core and shell' model that describes intraluminal thrombus formation; herein this model now describes the biology of blood escaping perpendicular to flow, forming a plug under a constant pressure head. The resultant architecture of the plug is a vaulted 'plug and cap' model, wherein pillars of platelets are formed which then serve to consolidate blood coagulation and red blood cell entrapment.

The data is complex and thus a mix of observational, descriptive and quantitative approaches; the data supports the results.

Of note, the authors a few times refer to puncture wounds being 'the oldest of scourges' of humans and thus drawing notions of evolutionary selection to their argument. The reviewer posits that in the light of evolutionary biology, the trauma and bleeding associated with child-birth (and menstruation) elicits the focus of selection pressure over getting poked by a thorn or piece of metal; in particular, note that metal came much later in the journey of human evolution. Death from a thorn is uncommon; death from childbirth still do this day is a selection pressure on the hemostatic system. The phrases and discussion points around this topic should be thoughtfully revised.

Rather than suggest further experimentation, rather a revised and expanded discussion that, rather than largely rephrasing the results, touches on a few points:

- 1) Scaling: do the authors posit that this phenomenon is conserved for punctures with increasing diameter, seeing that the platelet size remains the same. Would a similar structure be expected for a 100 micron, or 200 micron, etc size hole? The authors lightly suggest that this model could be extended to further the understanding (and management) of trauma – this should be expanded.
- 2) Translation from and to microfluidic models: several groups have been harkening on the development of 'bleeding chips'. Are principles uncovered in those studies supported by these findings, or rather does this study inform the design of improved models of puncture wound hemostatic plug formation. For instance: Cell Mol Bioeng. 2017 Feb;10(1):3-15; Cell Mol Bioeng. 2020 Aug 6;13(4):1-9; Nat Commun. 2018 Feb 6;9(1):509; J Thromb Haemost. 2018 May;16(5):973-983; Integr Biol. 2016 Aug 8;8(8):813-20.
- 3) The authors mention an increase or observation at least in leukocyte content (PMNs in particular) when the thrombin inhibitor is used. Does this tip the balance of innate immune response, thromboinflammation, etc? Is this a potential deleterious side effect of using DOACs that should be considered or even examined in patients?

Response to Referees, Communications Biology Submission, 12 January 2021

Brian Storrie, corresponding author

We appreciate the Reviewers' comments and have responded as per below and in the manuscript revision. These comments have helped produce a better manuscript. Our thanks to Reviewers for their time, effort and insights.

Blue highlight: Rebuttal

Yellow highlight: Action items edited or additional Supplemental Figure prepared

Referee expertise:

Referee #1: thrombosis, hemostasis, platelets

Referee #2: puncture models, assessment of vascular function in mice

Referee #3: platelets, thrombosis, hemostasis

Reviewers' comments:

Reviewer #1 (Remarks to the Author):

The manuscript by Rhee and colleagues is focused on an hemostatic response after a needle puncture of the jugular vein of adult mice. The authors describe a platelet plug morphology that was named "vaulted thrombus" which presents an inverted "U" shape and which is capped by platelets. The authors propose a new "Cap and Build paradigm that should have translational implications for bleeding control and hemostasis". While the authors used very powerful experimental approaches notably high quality imaging, and describe an unexpected and interesting platelet plug morphology, some results provided are not novel and quite expected, notably the sensitivity to anti-thrombotic agents or the presence and localization of pro-coagulant platelets. Moreover, the fact that they used only one approach to injure a vessel and only one vessel type is a major drawback of this study and the plug morphology described here cannot only based on this description represent a new paradigm.

Authors' answer: We present a model, a vehicle through which to interpret results. We freely acknowledge that this model is based upon the data from an idealized puncture wound. We present that there is compelling logic and compelling results that a sufficiently wide-open hole results in a vaulted thrombus, not a solid plug. We remind the Reviewer that as stated by an ISTH Committee on Recommendations and Guidelines, 31 January 2021, "Importantly, we bring the attention of the researchers to the fact that the relevant range of average wall shear rates in human arteries where clinically relevant arterial thrombosis occurs may fall as low as 100 to 200 s⁻¹, thus significantly overlapping with what are considered "venous" shear rates." In other words, what is true for venous could well have relevance for arterial. Our preliminary work in femoral artery does suggest that there may be carryover of puncture principles between vein and artery.

Major concerns:

1. The thrombus formation during an hemostatic response after penetrating injury has already been studied in the femoral artery (Welsh et al., 2016) and was described to present a "Core and Shell" structure. It seems essential that the authors study the hemostatic thrombus formation in other vessels after penetrating injuries to demonstrate the universality of a "Cap and Build" paradigm.

Authors' answer: Yes, as cited by the Reviewer, data is often interpreted within framework of the prevailing paradigms. The earlier intravital microscopy of the Furies in an endothelial layer damage situation pointed to a p-selectin positive Core. No one else has imaged within the full interior of a venous thrombus let alone an arterial example. As discussed immediately above, there is reason to expect that puncture principles could carry over between vein and artery. Future experiments will tell.

2. Can the authors explain why they proposed a new paradigm of “cap and build” while using the exact same model Tomaiuolo et al (2019) described a core and shell model? Is it not because the authors observed earlier time points compared to the other study (Tomaiuolo et al., 2019)? This would mean that there is no novel paradigm, but just an initial structure which does not exactly correspond to the core/shell model, but becomes a core/shell at late stages? We also wonder if the steps of thrombus formation are distinct for small (35-gauge) and big (23-gauge) injuries (Tomaiuolo et al., 2019)?

Authors' answer: The present work starts from the Tomaiuolo et al puncture model. Initial punctures were done by Dr. Stalker at Penn and later punctures by Ms. Kelly Ball at UAMS. The data summarized in Tomaiuolo et al. fail to image the interior of forming thrombus. Our approaches do exactly that. The Reviewer suggests that what is found by imaging the interior of the thrombus is simply minor differences in appearance. The Authors strongly disagree. The fundamental structuring is different and that difference in fundamental structuring arises because the principles of formation are different. These differences in formation are presented in some detail in the manuscript with respect to the 1 min time point. Pedestals and columns form. It is because the formation is different due to the geometry of the hole and how the exposed adventitia present that the resulting structure arises. The expectation of the Authors is that as the diameter of the puncture hole is reduced that eventually the experimental situation will approximate that of damage restricted to the endothelial layer and that as the hole diameter is increased that bleeding will not stop in the case of a sufficiently large hole. In clinical practice, stitches functions to restrict the size of larger wounds.

In brief, the reasoning of Tomaiuolo et al. is from a less complete and revealing data set that cannot reveal the outcomes established in the present work.

3. It seems that the authors believe that the vault-lining of degranulated platelets are procoagulant platelets. This is not strictly demonstrated and a better image quality would help. Could the authors use AnV or other markers for procoagulant platelets?

Authors' answer: Our assignment is based on morphology. As is summarized in Table 1, we used image sets for various purposes at different XY pixel size. **The Supplemental Data now include an image from an original 3 nm pixel size example showing at better resolution the apparent procoagulant platelets lining a vaulted area in a 5 min post puncture thrombus.**

4. Almost every experimental thrombosis model is dependent on anticoagulants, it is therefore not surprising that thrombosis in this model relies on dabigatran, and the link with the presence of PP in the vaulted region and the effect of an anticoagulant is far from being obvious.

Authors' answer: Yes, the Authors fully expected that dabigatran would affect thrombus structure; dabigatran at the concentration used prolongs bleeding time. The point here is how the structure is affected, an outcome that we submit is novel.

5. The authors propose a new paradigm, but it seems that in the plug they describe that most activated

platelets are in the center (core) and the loosely attached less activated platelets are on the edge (shell), which is not that different from the core and shell paradigm.

Authors' answer: The Authors draw the Reviewer's attention to one example, the fact that the vault surfaces, i.e., the column surfaces, are lined by the most highly morphologically activated platelets and the platelets in the interior of the columns are the least activated. This is a structure that would never be predicted by the Core and Shell model. Other examples are given the manuscript. In brief, a new paradigm, at least for the puncture hole case, is needed.

6. It would have been very informative to obtain some spatio-temporal information using intravital microscopy. The so-called pedestals seem to form at first, which is not very surprising as they expose the matrix where platelets adhere to.

Authors' answer: Yes, intravital microscopy could well be an important future approach. Our long-term hope is that newer imaging approaches developed for developmental biology purposes will permit such experiments with single cell resolution.

7. Figure 2: the quality of the EM images are very poor. They are probably good enough to localize RBC, but not informative enough to provide insight on platelets (level of activation, degranulation...). Fig 4B: darker staining is not a formal proof of stronger activation. Can the authors not focus and look at the granule content as a sign of activation level?

Authors' answer: The quality of the images cited by the Reviewer in Figure 2, presumably, in particular Figure 2C have little, if anything, to do with the quality of the EM done. Thrombi are large, approximately 400 to 500 microns in XY and 200 to 300 microns in Z. As shown in the new Table 1, the raw imaging was done at various XY pixel sizes ranging from 100 nm XY to 2 nm XY. A single plane might be 15,000 by 10,000 pixels. As developed in Table 1, choices were made to make this project possible. What is shown is the illustrative near "postage" stamp size outcome of that work. With respect to the specific point made by the Reviewer, a Supplemental Figure showing a subarea of a 3 nm XY pixel raw image of a 5 min post puncture thrombus is provided to illustrate that the choices made in our "coarse grained" analysis of thrombus structure were appropriate.

8. Movies were not provided.

Authors' answer: Movies will be uploaded with the revision. The journal wanted a reaction to the main body of the manuscript.

9. Can the authors explain why they observe bleeding with cangrelor while Tomaiuolo et al., 2019, had a much minor effect? According to text, in the current manuscript almost no platelet plug was detected, while the plug was clearly shown in the study of Tomaiuolo et al., 2019.

Authors' answer: The Reviewer is directed to Figure 7 of that paper. As shown in the "ii" row of that Figure, cangrelor at any of the needle sizes tested had a substantial effect on thrombus structure. In 60% of the 300 μm needle cases at 5-min post puncture, bleeding had not ceased, "iii". We would submit that there is no actual contradiction between our result and the full set of data presented by Tomaiuolo et al.

10. The authors wrote: "the "Cap and Build" paradigm could affect on the existing treatment protocols

because it indicates that enhanced bleeding cessation might be best achieved by local, external therapeutic application". This comment is surprising as covering the hole with sterile gauze piece is a standard procedure and not novel at all.

Authors' answer: The Reviewer's response suggests that our writing was not sufficiently explicit. Our intent was to suggest that administration of drug externally perhaps, through a drug laden gauze could be an efficacious route versus a standard intravenous delivery route. Localized delivery could have an advantage in minimizing secondary effects. **Wording in that section of the manuscript has been revised to make the point more clearly.**

11. Based on images in Fig 3 and supplement Fig 2, it seems that the hole could actually be filled. Again providing images over time or videos appears critical if the authors want to prove that in their new paradigm the first step of the process is not "filling the hole".

Authors' answer: Our data provides compelling evidence for a capping process that leads to bleeding cessation. In the 11 SBF-SEM data examples presented, we have never observed a solidly plugged hole. The three other examples as part of ongoing mutant studies show the same outcome. The fact that the now Table 2 quantitation of puncture hole diameter versus internal cavity diameter yields a significant difference by P value only at 5 min versus 1 min indicates in fact how open the cavity within the puncture hole is.

Minor comments:

1. Methods: what is the proof that the lesion exposes collagen and TF as proposed by the authors?

Authors' answer: **A new Supplemental Figure from 3 nm imaging has been added showing the presence of striated filaments lining the periphery of the puncture hole. The filaments display the expected morphological properties of collagen. A literature citation to expected TF exposure has been added.**

2. Methods: n=4 is a very low number of experiment for experimental thrombosis with the objective to determine injury size and detail the morphology of the plug? I would recommend to increase the number of experiments.

Authors' answer: In total for SBF-SEM, the n = 4 for 1 min, 5 for 5-min and 2 for 20 min in this study. Additional examples from individual WA-TEM examples are presented. These experiments produce both compelling and consistent qualitative and sufficiently coherent data to yield numerical trend data and at least in one case a significant P value as shown in what is now Table 2 of the revised manuscript. Since manuscript submission, one additional 5 min and two mutant thrombi have been analyzed as part of ongoing work. The outcomes from the outgoing work are entirely consistent with those of the present manuscript. These experiments began in the fall of 2016 and reached the stage of manuscript submission in mid January, 2021. That is bench work of 5 people per year for 4 years.

3. Figure 1: it is not easy to understand what we look at in Figure 1D

Authors' answer: **Wording has been editing to clarify that what is shown in Figure 1D is the pinned open jugular vein segment containing the thrombus in red (fluorescence Ab staining).**

4. Out of curiosity, the authors called the plug "thrombus": could the authors explain why they did not

use the term plug as they observe an hemostatic process, i.e. a biological response after injury of a healthy vessel wall?

Authors' answer: We used “thrombus” as what we perceived to be a neutral choice rather than “plug” which carries the connotation of a solid structure.

5. The fact that a thrombus rich in platelets contains region with RBC that are squeezed is not really surprising.

Authors' answer: Yes, the Authors realize that clot contraction occurs. The novel finding here is that the compression in the 20 min thrombi is concentrated in the central portion of the thrombus while the outer sheath of loosely adherent platelets is unaffected. **The revised manuscript has been edited to make this point clearer.**

6. The authors describe a Swiss cheese like interior with holes, but do they do not provide an idea of the number of holes per thrombus?

Authors' answer: In concept, this could be done. In ongoing work, we have segmented SBF-SEM thrombi and quantified the vault space in each of the individual thrombi, now ~15 in total. So, if we looked at the vault space in 3D, we should be able to count the holes. This analysis and number will be included in a future manuscript.

7. The authors wrote that the platelet aggregate formed after vessel rupture does not form a “plug”. A plug means connecting and if the thrombus formed is not similar to the one occurring after non-penetrating injuries, the aggregate is still sealing the breach by forming a connected structure, which could be considered as a plug.

Authors' answer: The puncture wound thrombi formed are as the title of the manuscript states “Vaulted”. We stay with what we see as the more neutral term of “thrombus” as versus “plug”, a structure described in all previous models as solid.

Reviewer #2 (Remarks to the Author):

Comments:

There are three major findings in the present study: 1) Bleeding cessation following venous puncture is due to extravascular capping of the puncture hole, rather than filling the hole to form a solid platelet plug; 2) the authors propose a novel principle of thrombus formation, that consists of pedestals and columns of vaulted structure thrombus with a central cavity. Thus, the suggested architecture of the thrombus structure in the present study dismisses the earlier concept of core and shell principle of thrombus formation; 3) the early steps in thrombus formation are sensitive to P2Y12 inhibition by cangrelor, and late steps to thrombin inhibition by dabigatran. The evidence in support of the Cap and Build principle of thrombus formation comes from the studies that employed correlative light and 3D electron microscopy together with supporting WA-TEM. The results of the study strongly support the Cap and Build hypothesis of thrombus formation following venous puncture. However, the authors may address the following queries.

1. Referring to the Introduction section, the statement “while extravascular thrombus capping was sensitive to the directly acting P2Y₁₂ ADP-receptor inhibitor, cangrelor” is not in accordance with the label of Fig 7F (Cap Dabigatran-sensitive). This may be looked into.

Authors’ answer: We appreciate the Reviewer calling this point to our attention to a point of wording on Figure 7F. The effect of cangrelor as shown here is greater than that of dabigatran at the indicated dose. We have edited the wording here to now state “Cangrelor sensitive Cap and Build, Build Dabigatran sensitive” as the effect of Dabigatran on build seems to be greater than that on capping.

2. Legend to the Table: “Cavity dimensions decrease with time”. If one compares the central cavity diameter between 1 and 5 min post-puncture, the values shown are not statistically significant. Then, how is it that cavity dimensions decrease with time?

Authors’ answer: The Table is setup to give emphasis to the comparison between puncture hole diameter and cavity diameter. That difference becomes significant at 5 min, $P = 0.006$ while at 1 min post puncture the P value is $P = 0.06$. This is a fact that actually emphasizes how little cavity closure there is at 1 min. **Additional wording has been added to emphasize this fact.** We have not actually made, but could have done measurements of accumulated platelet aggregate width or mass along the sides of the hole. Those measurements will be made in the future. There is actually platelet accumulation within the puncture hole.

3. The diameter of the central cavity of the vaulted structure is given in the Table. However, it is of geometrical interest to determine the height of columns and pedestals in order to understand the stability of the thrombus structure.

Authors’ answer: Yes, there are many additional numbers that are of interest and could be generated. We are in the process of doing further analysis as part of subsequent manuscript(s). In the opinion of the Authors, such efforts are beyond the necessary scope of the present manuscript.

4. “The height of the pedestals decreased with increasing distance from the puncture hole”. How was the height of pedestals determined? A graph illustrating the height of pedestals vs distance from the puncture hole will provide a quantitative expression of the results depicted in Fig 4B.

Authors’ answer: This statement is based on visual inspection of the 1500 to 2500 SBF-SEM slices for individual thrombi. The trend is sufficiently large to be visually obvious. The issue raised is peripheral to the present manuscript. As that is the case, the Authors would prefer not to take on the task now of defining how to do the quantification and the subsequent mathematical implementation of that definition.

5. In Fig 7B, the data with respect to the dose-related effect of dabigatran on bleeding time are shown. The authors should also provide the data relating to the effect of cangrelor on the bleeding time.

Authors’ answer: As stated in Methods, the cangrelor concentration used is the same as that used by Tomaiuolo et al. With a 30 gauge needle as reported by these authors, most animals bleed at 5 min post puncture.

6. Legend to Fig 7G: The schematic diagram shown in Fig 7G is not described in the figure legend.

Authors' answer: Our apologies, this shortcoming in manuscript preparation is corrected in the revised version.

7. It is described in the results section that "Intravascularly, the vaulted, platelet-aggregate defined spaces, were often open to the vessel lumen". What will be the impact of the blood flow (shear stress) inside the vessel on the thrombus structure?

Authors' answer: Mathematical analysis to answer the question of whether blood flow skews the shape of the thrombus is in progress and will be part of a separate manuscript.

8. Discussion section: "while a central cavity within the puncture hole was devoid of platelets and continuous with the vaulted intravascular vessel lumen." "more speculatively, a dynamic structure in which platelet movement/migration from intravascular to extravascular side might be occurring". There appears to be some discrepancies in the observed result and speculation about the platelet movement in the dynamic structure.

Authors' answer: Wording has been edited in the revised manuscript to bring clarity to these issues.

9. Will the size of injury influence the architecture of thrombus structure and the Cap and Build model of thrombus formation, suggested in this investigation?

Authors' answer: (Copy and Paste from a portion of response 2 to Reviewer 1) The expectation of the Authors is that as the diameter of the puncture hole is reduced that eventually the experimental situation will approximate that of damage restricted to the endothelial layer and that as the hole diameter is increased that bleeding will not stop in the case of a sufficiently large hole. In clinical practice, stitches function to restrict the size of larger wounds.

Reviewer #3 (Remarks to the Author):

This is an elegant descriptive study of bleeding and hemostatic plug formation at the site of a puncture wound. This adds to the now established 'core and shell' model that describes intraluminal thrombus formation; herein this model now describes the biology of blood escaping perpendicular to flow, forming a plug under a constant pressure head. The resultant architecture of the plug is a vaulted 'plug and cap' model, wherein pillars of platelets are formed which then serve to consolidate blood coagulation and red blood cell entrapment.

The data is complex and thus a mix of observational, descriptive and quantitative approaches; the data supports the results.

Of note, the authors a few times refer to puncture wounds being 'the oldest of scourges' of humans and thus drawing notions of evolutionary selection to their argument. The reviewer posits that in the light of evolutionary biology, the trauma and bleeding associated with child-birth (and menstruation) elicits the focus of selection pressure over getting poked by a thorn or piece of metal; in particular, note that metal came much later in the journey of human evolution. Death from a thorn is uncommon; death from

childbirth still do this day is a selection pressure on the hemostatic system. The phrases and discussion points around this topic should be thoughtfully revised.

Authors' answer: The Authors agree with the points raised regarding bleeding and the human population.

Rather than suggest further experimentation, rather a revised and expanded discussion that, rather than largely rephrasing the results, touches on a few points:

1) Scaling: do the authors posit that this phenomenon is conserved for punctures with increasing diameter, seeing that the platelet size remains the same. Would a similar structure be expected for a 100 micron, or 200 micron, etc size hole? The authors lightly suggest that this model could be extended to further the understanding (and management) of trauma – this should be expanded.

Authors' answer: (Copy and Paste from a portion of response 2 to Reviewers 1 and 2) The expectation of the Authors is that as the diameter of the puncture hole is reduced that eventually the experimental situation will approximate that of damage restricted to the endothelial layer and that as the hole diameter is increased that bleeding will not stop in the case of a sufficiently large hole. In clinical practice, stitches functions to restrict the size of larger wounds. Experimental tests of these expectations will follow with time and further effort.

2) Translation from and to microfluidic models: several groups have been harkening on the development of 'bleeding chips'. Are principles uncovered in those studies supported by these findings, or rather does this study inform the design of improved models of puncture wound hemostatic plug formation. For instance: Cell Mol Bioeng. 2017 Feb;10(1):3-15; Cell Mol Bioeng. 2020 Aug 6;13(4):1-9; Nat Commun. 2018 Feb 6;9(1):509; J Thromb Haemost. 2018 May;16(5):973-983; Integr Biol. 2016 Aug 8;8(8):813-20.

3) The authors mention an increase or observation at least in leukocyte content (PMNs in particular) when the thrombin inhibitor is used. Does this tip the balance of innate immune response, thromboinflammation, etc? Is this a potential deleterious side effect of using DOACs that should be considered or even examined in patients?

Authors' answer: There are multiple points raised here. A response to the query regarding microfluidic models is complicated. The data in the present work suggest that in vivo that granule in platelets results in the collapse of the granule membrane into the platelet surface plasma membrane. That is collapse is not observed in routine ex vivo experiments in thrombin, a strong agonist, is used. There are multiple explanations for this fact. One is temperature. The ex vivo experiments are typically done at room temperature to give a slower process that is easy and simply, for convenience. Washed platelets are used in comparison to the more complicated protein composition of plasma or PRP. I think we should solve this disconnect before trying to solve the more complicated problem of how to microfluidically model thrombus formation.

The thrombo-inflammation point raised is important. My personal opinion is that the approaches taken in the present study suggest that the actual effects of drugs be they P2Y12 inhibitors or DOACs in vivo is much more complicated than anticipated from past models and that as more research is done we will have better answers to questions raised. At the moment, all I urge is free thought and the application of

more revealing approaches. Discussion has been revised along the lines suggested above with new final wording.

Reviewers' comments:

Reviewer #1 (Remarks to the Author):

The authors acknowledge that their work is "preliminary" and we fully agree that it seems not correct to propose a new paradigm based only on one model with only one injury size and sometimes a low number of experiment. We firmly believe that reproducing (even partially) their main observation is the arterial bed is required to propose that the structure of the thrombus they describe results of a general mechanism and is valid for lesion occurring throughout the vascular bed.

Additional comments:

1. The Authors treated mice with cangrelor and dabigatran and indicated that these observations are novel. However, the effect of cangrelor was already reported for penetrating injuries where the thrombus is presenting a "Core-Shell" structure. The effect of dabigatran on the structure is novel but based on the literature, it is known that the impact of thrombin inhibitors on thrombus structure depends on the vessel and on the type of injury (Stalker et al. 2013; Welsh et al. 2016).
2. We fully agree with the reviewer that the shear flow conditions in arteries can be as low as 100 s⁻¹ (even lower) during the pulsatile cycle, but they can also reach values of 1,100 s⁻¹ in large arteries and even much higher in arterioles and in the microcirculation, which are never found in veins. In addition, this is not the only difference between the venous and arterial beds as the structure and composition of the vessel and its wall are also different. This further supports the fact that to propose a new paradigm and a universal model of thrombus formation after vessel injury, not one single vessel should be targeted with only one injury size.
3. It is not surprising to observe differences in thrombus ultrastructure in different vessels with different type of injuries and observing such differences are not sufficient to propose a new paradigm. For example, the ultrastructure of cremaster microvascular thrombi also present differences compared to the "Core and Shell" model (Courson et al. 2020), and the authors did not propose a paradigm shift.

Reviewer #2 (Remarks to the Author):

Comments:

There is a significant improvement in the above referred MS following the first revision. The authors have satisfactorily addressed all the queries that I had raised in my first review. I have no further comments on the revised MS.

Santosh K Mishra
Indian Veterinary Research Institute
Izatnagar, India

Response to Review

Two reviews were received. We are pleased that Reviewer 2 is completely satisfied with the manuscript following its recent revisions.

Reviewer 1 continues to have concerns regarding the need for a new puncture wound paradigm and the potential application of any principles to other vascular damage situations.

In response, we have made small, but significant changes in manuscript wording. For example, the word “Venous” has been inserted into the title to conclude the title on the phrase, “in a mouse puncture wound model”. Other similar changes have been made within the manuscript itself. The word “novel” has been deleted in several places within the manuscript in order to avoid unnecessary use. Other changes are highlighted in yellow in the current submission.

Furthermore, we have added an additional supplemental figure, Supplemental Figure 5, a wide area-TEM image of a cross section of a 20-min, post puncture, femoral artery thrombus. The femoral artery is a high flow situation. There are obvious similarities, even in this situation where repeat bleeding has occurred, to 20-min jugular vein, puncture wound thrombi. We present this as an illustrative example that model principles can be tested across system. We have limited interpretation to that in hopes of not over stating or over reacting in a situation where much additional data will come from future experiments.

We disagree with Reviewer 1 on the fundamental point of model and the need for a new paradigm upon which to frame the data. We present structural data on full puncture wound thrombi, their interior and periphery at an EM resolution sufficient to identify every platelet within the thrombus. These data are unprecedented. A major outcome of this work is clear evidence that the interior of the 5-min thrombus, a capped structure which stops bleeding, is **vaulted**. That is an outcome that cannot be explained by the current consensus Core and Shell model. The existing consensus simply piles platelets upon each other in first a solid Core and then a more porous Shell. This is simply not what happens within the interior of the puncture wound thrombus, 30 gauge needle. Our **Cap and Build** paradigm provides an explanation for this and other features of the data. In conclusion, it is the data that compel the formulation of a new paradigm. The exact limits over which that paradigm applies will then be revealed with further experimentation and time.

REVIEWERS' COMMENTS:

Reviewer #1 (Remarks to the Author):

no further request

Please check ref 26 as there are few errors: "sheer" instead of shear..."SS" instead of "SSC"

Rebuttal letter

July 15, 2021

Yes, as pointed out by Reviewer 1, there was a typo in Reference 26. That has been corrected.